# Ecological Network Optimization in Urban Central District Based on Complex Network Theory: A Case Study with the Urban Central District of Harbin

**DOI:** 10.3390/ijerph18041427

**Published:** 2021-02-03

**Authors:** Shuang Song, Dawei Xu, Shanshan Hu, Mengxi Shi

**Affiliations:** 1College of Landscape Architecture, Northeast Forestry University, Harbin 150000, China; songshuang0916@nefu.edu.cn (S.S.); S8565285@163.com (S.H.); mengxishi0608@163.com (M.S.); 2Key Lab for Garden Plant Germplasm Development & Landscape Eco-Restoration in Cold Regions of Heilongjiang Province, Harbin 150000, China

**Keywords:** landscape pattern characteristics, ecological network, complex network, edge-adding optimization, Harbin central district

## Abstract

Habitat destruction and declining ecosystem service levels caused by urban expansion have led to increased ecological risks in cities, and ecological network optimization has become the main way to resolve this contradiction. Here, we used landscape patterns, meteorological and hydrological data as data sources, applied the complex network theory, landscape ecology, and spatial analysis technology, a quantitative analysis of the current state of landscape pattern characteristics in the central district of Harbin was conducted. The minimum cumulative resistance was used to extract the ecological network of the study area. Optimized the ecological network by edge-adding of the complex network theory, compared the optimizing effects of different edge-adding strategies by using robustness analysis, and put forward an effective way to optimize the ecological network of the study area. The results demonstrate that: The ecological patches of Daowai, Xiangfang, Nangang, and other old districts in the study area are small in size, fewer in number, strongly fragmented, with a single external morphology, and high internal porosity. While the ecological patches in the new districts of Songbei, Hulan, and Acheng have a relatively good foundation. And ecological network connectivity in the study area is generally poor, the ecological corridors are relatively sparse and scattered, the connections between various ecological sources of the corridors are not close. Comparing different edge-adding strategies of complex network theory, the low-degree-first strategy has the most outstanding performance in the robustness test. The low-degree-first strategy was used to optimize the ecological network of the study area, 43 ecological corridors are added. After the optimization, the large and the small ecological corridors are evenly distributed to form a complete network, the optimized ecological network will be significantly more connected, resilient, and resistant to interference, the ecological flow transmission will be more efficient.

## 1. Introduction

Under the background of rapid urbanization, excessive utilization and development of urban resources and environment lead to deterioration of the regional ecological environment and obstruction of ecosystem service function, which seriously affects regional landscape pattern and sustainable development [1,2,3,4]. With urban sustainable development as the goal, ecological network optimization applies certain techniques and methods to combine landscape structure, ecological process, and ecological function to layout various landscape elements in the region, obtain a multi-objective, multi-level and multi-category landscape spatial network composed of points, lines and planes [5,6,7]. Optimization of the urban ecological network is an important means to stabilize the ecosystem, strengthen the association and interaction among natural patches, maintain the normal play of ecosystem service functions, and enhance the natural carrying capacity [8,9]. In 1986, two conceptual models, “irreplaceable pattern” and “segregation between clusters”, were important methods of early landscape ecological pattern optimization [10,11,12,13]. With the development of remote sensing and GIS technologies, the optimization methods of ecological network based on landscape pattern indexes and a series of models have been proposed successively, which have laid the research foundation for improving the quality of urban ecological network [14,15,16].

An ecological network can be understood as a large and complex system consisting of three landscape elements: regional ecological sources, corridors, and nodes [17]. Complex network theory is a new perspective and method to study complex systems by analyzing the topology of the interrelated roles of individuals, and optimizing ecological network based on complex network is one of the frontier directions in this field [6]. Complex network theory comes from graph theory, combines systems science, sociology, and other theories [18,19,20,21]. Complex network is a topological structure that focuses on the correlation between individuals in the system and is the basis of understanding the nature and function of complex systems [22], with the aim of understanding the function of various real network systems and guiding the design and optimization of the actual network system [18]. At the end of the 20th century, the complex network attracted the attention of many researchers due to the publication of two important academic papers on small-world networks and scale-free networks [23,24], various complex network topology models have been proposed, and widely applied to social network, expert network, aviation network, transportation network and other fields [24,25]. However, until now there is still no unified and comprehensive definition for complex systems, existing research believes that small-world characteristics, self-similar characteristics, self-organizing features can be called complex network, the meaning of complex network has two main points: a summary of topology abstract from a large number of real systems; a network between the regular network and random network [19]. The first study that applied complex network theory to ecological network appeared in 2008, but the network optimization in the study was still based on the traditional landscape pattern theory and rarely involved complex network theory [20]. Since then, the research on ecological network optimization using complex network theory has gradually advanced, such as: Analyzing desert oasis ecological network based on complex network theory and introducing it into ecological network research to analyze network topology and robustness characteristics [6]. The low-degree-first strategy was applied to optimize the landscape ecological network, and the connection robustness was used to evaluate the connection ability of the optimized ecological network after suffering damage to [26]. At present, the application of complex network theory in ecological network mainly has two ways: One is to analyze the topological structure and robustness of ecological network [25]. The other is to optimize the landscape ecological network through specific parameters and evaluate the connection ability of the optimized ecological network by connection robustness [26]. Existing studies have played a role in expanding the application of complex network theory in landscape ecology, but have failed to fully demonstrate the optimization effect of the theory on the ecological network, the object of previous studies mainly focus on desertification areas, with limited guidance for the optimization of the ecological network in urban landscapes where the contradictions between natural ecosystems and urban construction are becoming increasingly prominent. And the scope of previous studies is mostly large-scale regions such as cities or counties, the research on the optimization of landscape patterns of the central urban area is rare. The reasonable landscape pattern of the central urban area has an important influence and role in improving the regional ecological environment and the living environment and quality of life.

This study breaks through the previous situation of using complex network theory to optimize the landscape ecological network in large scale desertification areas and selects Harbin, a northern provincial capital city with increasingly serious ecological environment problems, to conduct a study on the optimization of its central urban landscape ecological network using the principles of landscape ecology and complex network theory, aiming to build an urban landscape ecological network pattern with complete ecological functions and connected ecological network. Thus, the aims of this study were to: (1) start from land use, use GIS spatial analysis techniques for remote sensing interpretation of the study area, quantitative study of Harbin central urban ecological landscape pattern status; (2) complete the ecological network extraction of the study area with minimum cumulative resistance (MCR) model; (3) optimize the ecological network by edge-adding of the complex network theory, compare the optimizing effects of different edge-adding strategies by using robustness analysis, and propose an effective way to optimize the regional ecology in the study area. To provide a scientific basis and decision support for ecological network connectivity and good ecological environment restoration in the study area.

## 2. Materials and Methods

### 2.1. Study Area

Harbin is located in the south-central part of Heilongjiang Province, China, between 44°04′–46°40′ N and 125°42′–130°10′ E, is the provincial capital of Heilongjiang Province, the central city northeast, and an important manufacturing base in China [27]. The site belongs to the mid-temperate continental monsoon climate, with an average annual temperature of 3.6 °C, long severe winters, and short cool summers, average annual precipitation is 569.1 mm, main precipitation months being from June to September, accounting for about 60% of the annual precipitation, main snow month is from November to January of the following year [28]. The overall topography is high in the east and low in the west, with mountains and hills predominating in the east and plains in the west. The specific scope of its central district includes Daoli district, Daowai district, Nangang district, Xiangfang district, Pingfang district, Songbei district’s administrative district, Hulan district, and Acheng district part of the area, with a total area of about 4187 km^2^ (Figure 1).

From the second round of urban master plan of Harbin in 1981 with Nangang district and Daoli district as the core clustered spreading development to the fifth round of urban master plan focusing on the development of the Hanan new city, Qunli new district, and Haxi new district, the city scale is rapidly expanding. According to the “Harbin City Urban Master Plan (2011–2020)” (revised draft in 2017), the resident population in the central district is controlled at 7 million in 2020, with a non-agricultural population of 6.16 million and an urban level of 88%. With the rapid urbanization process and its in-depth implementation of the revitalization of the old industrial base of Northeast China’s urban development strategy, the construction of the central district of Harbin will continue to face many ecological and environmental problems [28]: A large amount of agricultural land in the suburbs and ecological land is constantly encroached upon, the agricultural population continues to move to the city, the habitat of living creatures is getting smaller and gradually eaten away, biodiversity loss, a large number of shrinking wetlands and lakes, some important ecological corridors have been partitioned, separated and encroached, resulting in discontinuity of their ecological functions and deterioration of ecological quality. For this reason, the Harbin municipal government has emphasized the importance and necessity of strengthening urban ecological planning in each round of urban master planning. In the “Harbin City Master Plan (2004–2020)”, it is emphasized that the urban ecological planning layout is formed into “one river, two rivers, three ditches, and four lakes” as the link of the urban water system, south river and north island, east mountain and west lake as the greening focus. It also emphasizes the importance of water systems such as the Songhuajiang and Hulan River and the Sun Island scenic spot of the ecological pattern of the city. In the “Harbin City Urban Master Plan (2011–2020)” (revised draft in 2017), it is proposed that by 2020, a green space ecological network system combining points, lines and surfaces will be formed with a “ring-wedge-shaped radial-grid”, and the green space rate, per capital green space and other indicators have specific requirements, and focus on the emphasis on the layout of green space in the central city. The above planning and green space indexes reflect the characteristics of Harbin’s urban ecological planning and the importance attached to ecological construction.

At present, the ecological elements in the study area are mainly waters, scenic spots, city parks, and protection forests (Figure 2). The waters include the three main rivers of Songhuajiang, Ashi river, and Hulan river and their wetland systems on both sides of the rivers, and all three rivers are affected and blocked to different degrees by their tributaries. The scenic spots mainly include Tiane lake, Tianheng rake, Taiping rake, Sun Island scenic spot, and Changling lake scenic spot. The natural system is relatively complete, but the area is small and the degree of ecological function is limited. There are five city parks with an area larger than 1 km^2^ in the current study area, which occupy an important position in the landscape ecological network pattern of Harbin. Among them, the urban parks in Xiangfang district, and Pingfang district are more concentrated, and the ecological corridors are better protected, and the parks are connected by corridors to form complete ecological patches, while the rest of the parks in the study area are scattered within the distribution, with small areas and serious problems of corridor discontinuity. There are more than 10 pieces of protection forests in the study area, which can be divided into urban forests and protected greenfield. The current distribution of protection forests is relatively scattered, mostly scattered along the railroad line and urban-rural combination, which does not form an ecological network system, and there is a lack of protected greenfield along the railroad line and in the chemical industry areas of Hadong and Haxi. In general, Harbin’s urban ecological network space lacks systematic and low connectivity, and the fragmented ecological land leads to a continuous decrease in ecological quality.

### 2.2. Data Sources

Our data mainly included 2020 Landsat RS images (spatial resolution 30 m, source http://www.gscloud.cn), through a series of pre-processing (radiation calibration, atmospheric correction, fusion, and cutting), and human-computer interaction is applied to decode the land use data in the study area. And normalized vegetation index (NDVI) and modified normalized difference water index (MNDWI) data (source http://www.resdc.cn/), elevation (DEM) and slope data (source http://www.gscloud.cn), road density and rail density data (source http://www.Openstreetmap.org/).

### 2.3. Methods

#### 2.3.1. Landscape Pattern Analysis

The landscape pattern indexes describe the complexity of the type and arrangement of landscape patches through information on patch shape, size, number, and spatial combination. A single index cannot be used to comprehensively analyze the features of landscape ecological patterns, and there is often a correlation between landscape indexes [29,30,31]. Therefore, referring to related research results and combining them with the actual situation of the study area, this study selected the number of patches (NP), class area (CA), percentage of landscape (PLAND), largest patch index (LPI), Landscape shape index (LSI), and patch cohesion index (COHESION), quantitatively studied the landscape pattern characteristics of the study area based on the above six indexes [32,33]. The formulae for calculating indexes and the ecological significance are referred to in the Fragstats4.2 manual.

#### 2.3.2. Extraction of Ecological Network

The extraction process of the ecological network mainly includes determining the ecological sources, constructing ecological cumulative resistance surface, extracting ecological corridors and ecological nodes, the urban ecological patches are the carrier of the urban ecological sources [34,35,36]. The ecological source of this study refers to the ecological patch such as green space, scenic spots, waters, and woodland that has important protection value for urban landscape ecology. In this study, patch area, average NDVI/MNDWI value, and patch shape index were used to evaluate the ecological patch importance (Q), and the first 60% patches were selected as the ecological sources [25,26]. Comprehensively considering the land use type, terrain, MNDWI, NDVI, road density, and rail density (Table A1), the MCR model was used to construct the ecological cumulative resistance surface of the study area. The general form of the MCR model is as follows [37]:(1)$$R_{mc}=f_{\min}\sum D_{ij}R_{i}^{}$$
where *R_mc_* is the minimum cumulative resistance, *f_min_* is an unknown negative function, *D_ij_* is the spatial distance from the source to landscape unit, *R_i_* is the resistance coefficient of the landscape unit to the propagation and diffusion process.

The ecological corridor is the minimum cost path of landscape ecological flow from one ecological source to another through the ecological cumulative resistance surface [38], this study used the minimum cost method to extract ecological corridors. The ecological node is the key point in the ecological network, the study obtained ecological nodes by extracting the spatial distribution centroids of ecological sources [28]. The ecological source, ecological corridor, and ecological node extracted by the above method were superimposed on the same layer to form the ecological network of the central district of Harbin.

#### 2.3.3. Optimization of Ecological Network

As complex network, the optimization of the topology of ecological network seriously affects the smooth flow of materials and energy. There are four main methods to optimize the topology of complex network theory: edge-deleting, edge-adding, edge-reconnecting, and edge-orienting [39]. The edges of the ecological network are actually ecological corridors with the minimum cost of biological flow connecting the ecological sources. The deletion of edges is the destruction of the ecological corridors, so the deletion and reconnection mechanisms are not appropriate for the optimization of the ecological network. Because the ecological network is an undirected topology, edge-orienting is also not appropriate [40,41]. The edge-adding strategy is optimized in the following four specific ways [42,43]:

(1) Random addition (RA): edges are added between two randomly disconnected ecological nodes in the ecological network. (2) Low-degree-first (LDF): according to the ecological node degree value, edges are added between the two nodes with the smallest and unconnected ecological node degree value. (3) Low-betweenness-first (LBF): according to the ecological node betweenness value, edges are added between the two nodes with the smallest and unconnected ecological node betweenness value. (4) Shortcut for maximum betweenness (SMB): select the node with the largest betweenness of the ecological network, edges connected to the node are sorted by betweenness from small to large, select the two nodes with the smallest betweenness and the other node is not connected to increase edges. In this study, the ecological network of the study area was optimized by using the above four edge-adding strategies, and the number of edge-adding was set to 30% of the existing ecological corridor number. Among them, the edge-adding strategy contains two important statistical parameters in complex network theory: degree and betweenness.

The degree is the primary indicator of node importance and is the number of nodes directly connected to that node, the greater the node degree is, the more important this node is [40]. The average degree is the average value of node degree in the network, which is an important indicator reflecting the structure of the ecological network. The equation of average degree is as follows:(2)$$k=\frac{1}{N}\sum_{t=1}^{N}k_{i}$$
where *k* is the average degree of the network, *N* is the total number of nodes, *k_i_* is the node degree of node *i*.

Betweenness, which is divided into node betweenness and edge betweenness, is an index reflecting the influence of nodes (edges) in the network. It can be understood that in the control of biological flow transmission if the node betweenness values larger, it is more important, these nodes are removed after the biological flow transmission loss is greater. If many shortest paths between two non-adjacent nodes *i* and *j* in the network pass through a node, it means that the node is very important. The equation of node betweenness is as follows [41]:(3)$$B_{v}=\sum_{i\neq j\in V}\frac{\delta_{ij}\left(v\right)}{\delta_{ij}}$$
where *B_v_* is the betweenness of node *v*, *V* is the set of all nodes in the network, $\delta_{ij}(v)$ is the number of nodes *v* passing through the shortest path of node *i* and *j*, $\delta_{ij}$ is the shortest path number between nodes *i* and *j*.

To characterize the properties of edges, similar to node betweenness, the concept of edge betweenness is introduced. The larger the betweenness of the edge, the more the amount of data through the edge in the process of information transmission, the blockage will occur to the transmission process. The equation of edge betweenness is as follows [40,41]:(4)$$B_{l}=\sum_{i\neq j\in V}\frac{\delta_{ij}(l)}{\delta_{ij}}$$

#### 2.3.4. Optimization Effect Test of Ecological Network

In this study, the robustness was used to test the optimizing effect of different edge-adding strategies. Robustness refers to the ability of the network to maintain its function after some nodes (edges) fail after being attacked by the outside world which is divided into connection robustness and recovery robustness [44]. Connection robustness refers to the ability of the network to maintain its normal structure and function after the network structure is destroyed (removing some nodes or edges), recovery robustness refers to the recovery ability after the network structure is destroyed [39]. The connection robustness equation is as follows:(5)$$R=\frac{C}{N-N_{r}}$$
where *C* is the maximum number of nodes in the network connecting the subgraph after removing some nodes, *N_r_* is the number of nodes to be removed.

Recovery robustness can be measured by edge recovery robustness and node recovery robustness. In this study, they represent the recovery ability of corridors and nodes in ecological networks after being destroyed [45]. The equations of edge and node recovery robustness are as follows:(6)$$\begin{array}{l} D=1-\frac{N_{r}-N_{d}}{N}\\ E=1-\frac{M_{r}-M_{e}}{M} \end{array}$$
where *N_d_* is the number of nodes recovered by the network after removing some nodes, *M_r_* is the number of edges removed, *M_e_* is the number of edges restored, *M* is the total number of edges.

In this study, the changes of connection robustness, node and edge recovery robustness under random attack, and intentional attack before and after network optimization were analyzed to compare the optimizing effects of different edge-adding strategies. Random attack means removing some nodes randomly of the network, intentional attack means removing some nodes with the maximum degree and their corresponding edges of the network, simultaneously. It should be noted that all potential ecological corridors are extremely important to the ecologically fragile urban environment. Therefore, when this study uses robustness comparison to test different optimization strategies, it is assumed that the edge weights of the actual ecological network are equal, and since the flow of ecological flow in the ecological corridor is bidirectional, the ecological network is regarded as a directionless and weightless network in this study.

## 3. Results and Discussion

### 3.1. Analysis of Landscape Pattern of Ecological Patches in the Study Area

The ecosystem in the central urban area of Harbin is mainly composed of an ecological patch network. To deepen the understanding of the current situation of ecological patches, this study made use of the above landscape indexes to deepen the analysis of ecological patch problems in each district of Harbin central district (Table 1). According to the NP and CA indexes, there are 198 ecological patches with a total area of 64,999.71 hm^2^ in the study area, the ecological patches are small in size and fewer in number in Daowai, Xiangfang, Nangang, and Pingfang districts, and large in size and more numerous in Songbei, Hulan, and Acheng districts. In other words, the closer the ecological patches are to the urban center, the more obvious the distribution of ecological patches is in the form of points, and the farther away from the urban center, the more likely the ecological patches are to appear in the form of surfaces, the main reason for this phenomenon is that the development of the city is too random and does not follow the original ecological system. The largest PLAND index of 21.8475% is found in Songbei district, where large intact ecological patches existed, and similarly, the PLAND index was larger in Daoli, Daowai, Songbei, and Acheng districts. The LPI indexes of ecological patches in Daoli, Xiangfang, Nangang, and Pingfang districts are much smaller than those in Daowai, Songbei, Hulan, and Acheng districts, and the fragmentation of ecological patch pattern of these areas is more serious, which makes it more necessary to establish an ecological network to organize and integrate the ecological elements in the region. The LSI indexes of Songbei and Acheng Districts are significantly higher than those of old urban areas such as Nangang, Xiangfang, and Daowai districts, reflecting the higher proportion of natural ecological patches in these two districts and the more complex shape of natural patches. In contrast, the LSI indexes of old urban areas are low, mostly artificial ecological patches with a single shape, and the ecological patches have fewer opportunities to exchange materials and energy with the surrounding environment, and their overall contribution to the urban ecological environment is low. The COHESION indexes are low in Xiangfang, Nangang, and Pingfang districts, indicating that the patches have high internal porosity and low connectivity, and these three districts are influenced by more external factors, leading to more serious damage to the internal connectivity of ecological patches, and attention should be paid to strengthening corridor connectivity among ecological patches.

Combining the results of the above landscape indexes analysis, from the perspective of urban ecological planning, Daoli, Daowai, and Nangang districts undertake most of the city’s traffic, socio-economic and other activities and urgently need the regulation of ecological factors. At present, the ecological patches in these areas are sparse and severely fragmented, and the ecological network is weak and unformed, which needs to be strengthened urgently. Xiangfang and Pingfang districts are mainly responsible for urban housing and economic development zones, but the quality of patches in these districts is not high and the internal connectivity of similar patches is poor, which is not conducive to the establishment of urban ecosystems and the connection of urban ecological networks. The Songbei, Hulan, and Acheng districts are separated from the other five districts, which are far away from the urban center, and they are the flowing areas of the Songhuajiang and the Hulan River, the water system is rich and self-contained, and the original ecological patch foundation is good. However, these three districts are the main direction of urban development and expansion in Harbin, building land has multiplied in recent years, and more attention should be paid to the sustainable development of the ecological system.

### 3.2. Construction of Ecological Network in the Study Area

#### 3.2.1. Identification of Ecological Sourcess

According to the evaluation results of the importance of ecological patches, the first 60% patches were selected as the ecological sources, 119 ecological sources were identified. The results of the ecological patches importance (Q) statistics are shown in (Figure 3). Fewer ecological patches have higher Q values, patches with lower values occupy the majority. The Q value is the highest of 0.95, with 10 patches between 0.3–0.9, and the Q value of most ecological patches is between 0.05 and 0.75.

The total area of ecological sources is 42,160.16 hm^2^, accounting for 10.07% of the total area of the study area, which is mainly divided into three categories: Watershed ecological sources (canal, lake, reservoir, marsh), greenfield ecological sources (high coverage grassland, medium coverage grassland), and woodland ecological sources (woodland, bush, sparse forest). Among them, the watershed ecological source is the largest source type, the total area is 24,678.24 hm^2^, including 33 patches, concentrated distribution along the Songhuajiang and the Hulan river, the western and southern study area also has sporadic distribution. The total area of woodland ecological sources is 10,547.32 hm^2^, with 52 patches, which are concentrated in Acheng district and scattered in Songbei district. The total area of greenfield ecological sources is 6934.60 hm^2^, with 34 patches, mostly distributed among Songbei and Acheng districts, mainly including most of the grassland in Acheng district, Sun Island Scenic Spot, and Changling Lake Scenic Spot. On the whole, the ecological sources of Daowai, Xiangfang, Nangang, and other old districts in the study area are fewer in number, smaller in area, showing isolated distribution, less connected to other ecological sources, and lower in biodiversity and stability within the isolated ecological sources (Figure 4).

#### 3.2.2. Extraction of Ecological Corridors

The total resistance consumption surface of the study area was obtained by overlaying the resistance consumption surface of land use type, terrain, MNDWI, NDVI, road density, and rail density data (Figure 5). The building land and road network in the central district of Harbin occupy the absolute area and density, which have high resistance to the accumulation of biological flow, and are not conducive to species, energy migration, and diffusion.

Based on the ecological source and the minimum cumulative resistance surface, 262 ecological corridors were extracted in the study area, and 142 ecological corridors were obtained after the overlapping paths were processed. The ecological corridor from the study area mainly includes the Songhuajiang ecological corridor, the Hulan river ecological corridor, the Ashe river ecological corridor, the Majiagou river ecological corridor, the Xigou ecological corridor, and so on, mainly connecting the Songhuajiang, the Changling lake scenic spot, the botanical garden, the Yunliang river scenic spot, the Eurasian window, and other important ecological sources. The number of watershed ecological sources and greenfield ecological sources that are crossed by ecological corridors is relatively high, while the woodland ecological sources are low, and among all types of ecological sources, there exist phenomena that are not crossed by ecological corridors. From the distribution of corridors, we found that there are few easily identifiable ecological corridors in the study area, some of them are blocked by urban construction, and the ecological network is generally poorly connected. The extracted ecological corridors in the study area are sparsely scattered, the corridors in Hulan, Xiangfang, and Nangang districts are spatially shrunken, various ecological sources are not closely connected through the corridors, some ecological sources only have a single corridor connection with the surrounding ecological sources. Only the ecological corridors in Acheng and Songbei districts have a good degree of circularity, the ecological network skeleton of urban ecological planning is basically maintained, but the local ecological network structure is problematic (Figure 6).

### 3.3. Robustness Analysis of Ecological Network before and after Optimization

#### 3.3.1. Robustness Analysis of Ecological Network before Optimization

The initial value of the network connectivity robustness index before the optimization is 0.70, which has poor connectivity. Under intentional attack, the connectivity robustness of the network decreases dramatically. When the number of removed nodes reaches 17, the connectivity robustness index of the network is less than 0.1, at this time, the network structure has been completely destroyed and the connectivity function of the network is almost completely lost. The connection robustness index is below 0.1 when 186 nodes are removed under random attack (Figure 7a).

As the nodes removed increases, the node recovery robustness of the network under both attacks tends to decrease, with the overall slope of the decreasing curve for the random attack being smaller than that of the intentional attack. Under intentional attack when the removed nodes are less than 19, the destroyed nodes in the network can be entirely recovered. When the removed nodes are less than 24, the destroyed edges could be fully recovered. Under random attack, the destroyed nodes in the network can be entirely recovered when the removed nodes are less than 20. When the removed nodes are less than 34, the destroyed edges can be fully recovered. With the increase of the removed nodes, the recovery robustness continues to decrease under both attacks (Figure 6).

#### 3.3.2. Robustness Analysis of Ecological Network after Optimization

Four edge-adding strategies of complex network theory were used to optimize the ecological network of the study area, and the number of edge-adding was all set to 30% of the existing number, reaching 185. After optimization, the connection robustness varies with node removal as follows (Figure 8). The initial value of the connection robustness index after the optimization of the four edge-adding strategies is 0.72, 0.98, 0.93, and 0.92, respectively, the LDF and LBF are significantly improved compared with before optimization. Under intentional attack, the number of nodes removed by the four edge-increasing strategies reaches 22, 68, 18, and 18, the connectivity robustness index is below 0.1, and the network connectivity is completely destroyed. Under random attack, when the connection robustness of the network is less than 0.1, the number of nodes removed by the four edge-adding strategies is 218, 228, 235, and 196, respectively. Compared with before optimization, the effect of LDF optimization is the best, the initial value of connection robustness is the highest, and the decline rate is the slowest, followed by LBF.

After optimization, the recovery robustness varies with node removal as follows (Figure 9). Under intentional attack, the destroyed nodes in the network can be entirely recovered when less than 28, 26, 18, and 18 nodes are removed, the destroyed edges in the network can be entirely recovered when less than 25, 36, 34, and 26 points are removed by the four strategies. RA is the best for point recovery robustness optimization, followed by LDF, LBF and SMB are less robust for point recovery than before optimization. LDF is the best for edge recovery robustness optimization, and the remaining three strategies are the second-best for edge recovery robustness optimization.

Under random attack, the destroyed nodes in the network can be entirely recovered when less than 36, 48, 46, and 37 nodes are removed, and the destroyed edges in the network can be entirely recovered when less than 28, 42, 44, and 28 points are removed by the four strategies. LDF has the best optimization for node recovery robustness, the other three strategies have different degrees of improvement over the pre-optimization. LBF has the best optimization for edge recovery robustness, followed by the LDF, RA and SMB have a decrease in edge recovery robustness over the pre-optimization network.

In summary, the network connectivity robustness index before the optimization is 0.7, the network is completely destroyed when 45 nodes are removed (intentional attack), 186 nodes are removed (random attack). Among the four edge-adding strategies, the robustness of network connection optimized by LDF enhancement is the most significantly improved than that before optimization. The initial index of connection robustness rises to 0.98, the network can be completely destroyed when 68 nodes are removed (intentional attack) and 228 nodes are removed (random attack).

Before optimization, the destroyed nodes in the network with the number of removed nodes less than 19 (intentional attack) and 20 (random attack) can be entirely recovered, and the destroyed edges in the network with the number of removed nodes less than 19 (intentional attack) and 34 (random attack) can be entirely recovered. Comparing the recovery robustness after optimization of different strategies, RA has the best point recovery robustness optimization under intentional attack (removed nodes is improved to 28). LDF has the best point recovery robustness under random attack (removed nodes is improved to 48). LDF is the best in optimizing the recovery robustness of edges under intentional attack (removed nodes is improved to 36). LBF is the best in optimizing the recovery robustness of edge under random attack (removed nodes is improved to 44). Therefore, LDF performs best for the optimization of the network. The use of LDF to optimize the ecological network in the central district of Harbin not only enhances the material circulation capacity of the ecological network but also enhances the resistance to damage and recovery capacity of the nodes and corridors of the ecological network.

### 3.4. LDF-Based Ecological Network Optimization in the Study Area

Before optimization, the average degree of the ecological network in the study area is 2.4196, the maximum degree is 5, and the number of nodes is 5, distributed in the Songbei, Daowai, and Acheng districts. The largest number of ecological nodes with degree 2, 46 in total, and 5 ecological nodes with degree 0 (Figure 10). The node degrees of some woodland ecological sources and greenfield ecological sources in the ecological source are large in the ecological network of the study area, the node degrees of watershed ecological sources are relatively low, or even 0. The node degree of ecological sources in the old districts such as Daoli and Dawai is generally low because there are fewer ecological sources of the study area and the number of ecological corridors linked to them is relatively low. The higher the node degree of an ecological source, the higher the external connectivity of that ecological source. Ecological sources with lower or 0 nodal degree are the focus of LDF to enhance ecological corridor linkages.

Combining the current landscape pattern of the study area, adopting LDF to optimize the ecological network, the average degree of the ecological network is 2.8839, the maximum degree is 5, and the number of nodes grows to 7. The number of ecological nodes with degree 3 is the highest, with 50. There are 2 ecological nodes with degree 1, and there are no ecological nodes with degree 0 (Figure 11). From the optimized ecological network of the study area, we can see that Songbei and Acheng form a circular plus radial ecological networks, and the ecological network of other districts have a better degree of connectivity. Large ecological corridors constitute the backbone of the ecological network and basically cover the whole area. Small ecological corridors are evenly distributed among the gaps of the network formed by large ecological corridors and are an effective complement to the whole ecological network. Overall, the optimized ecological network of the study area forms a stable ecological pattern with primary and secondary importance, focus and clarity (Figure 12).

In traditional urban planning, there is often a lack of in-depth research on the spatial distribution of potential ecological corridors and ecological nodes in the region, resulting in neglecting the protection and utilization of potential ecological corridors and ecological nodes in urban construction, which largely undermines the regional ecological environment security. The study discards the traditional urban planning ideas, fully considers the regional landscape ecological pattern, takes the current ecological spatial distribution as the basis, introduces the edge-adding optimization strategy of complex network theory to the urban ecological construction research, and quantitatively compares the best method of urban ecological network optimization to further scientificity the planning procedures and modes of ecological planning establishment and optimization. And uses the ecological patches of key points for minimal intervention to achieve the maximum ecological transformation of urban, linking the limited ecological resources, it provides a reference for the comprehensive deployment and arrangement of various land spaces in urban planning.

The optimized ecological network will be significantly more connected, resilient, and resistant to interference, the ecological flow transmission will be more efficient. The results of the study provide ideas of optimizing the ecological network construction in Harbin, strengthening the stability of the urban landscape ecological network, improving the overall ecological effectiveness within the urban ecological landscape, and realizing the sustainable development of urban ecology. At the same time, it has an important reference significance of the planning and construction of the ecological network in other cities. Besides, this study uses the undirected and unweighted analysis method of complex network theory for ecological network topology analysis, but the development of urbanization will make the importance of regional ecological corridors different, and this ecological network is undirected and weighted, which leads to the accuracy of the research results need to be further improved. Therefore, in the actual network optimization also on this basis to make appropriate adjustments, and the optimization of this undirected and weighted urban ecological network is the future direction of complex network theory research.

## 4. Conclusions

(1) There are 198 ecological patches with a total area of 64,999.71 hm^2^ in the study area. The ecological patches of Daowai, Xiangfang, Nangang, and other old districts in the study area are small in size, fewer in number, strongly fragmented, with a single external morphology and high internal porosity, there is an urgent need to establish an ecological network to organize and integrate the ecological patches in the region. While the ecological patches in the new districts of Songbei, Hulan, and Acheng districts have a relatively good foundation, however, these three districts are the main direction of urban development and expansion in Harbin, building land has multiplied in recent years, and more attention should be paid to the sustainable development of the ecological system.

(2) The ecological network of the study area constructed by MCR consists of 119 ecological sources and 142 ecological corridors. The number of watershed ecological sources and greenfield ecological sources that are crossed by ecological corridors is relatively high, while the woodland ecological sources are low, and among all types of ecological sources, there exist phenomena that are not crossed by ecological corridors. There are few easily identifiable ecological corridors in the study area and they are sparsely distributed, some of them are blocked by urban construction, the ecological network connectivity is generally poor. The ecological corridors in Hulan, Xiangfang, and Nangang districts are shrinking, only the ecological corridors in Acheng and Songbei districts have a good degree of circularity, but the local ecological network structure is problematic.

(3) The optimization of the ecological network of the study area is performed by using the edge-adding optimization strategy of complex network theory, and the optimizing effects of four edge-adding strategies are compared by the changes of connection robustness and recovery robustness of ecological network with different attack methods. The LDF strategy has the best overall performance and the most significant improvement in network connectivity robustness of the optimized network compared to the pre-optimization, it has the best recovery robustness for nodes under random attack and the best recovery robustness for edges under intentional attack. In this study, the ecological network was optimized by adding 43 ecological corridors using LDF, after optimization Songbei and Acheng form a circular plus radial ecological network, and the ecological network of other districts have a better degree of connectivity. The optimized ecological network will be significantly more connected, resilient, and resistant to interference, the ecological flow transmission will be more efficient

## Figures and Tables

**Figure 1 ijerph-18-01427-f001:**
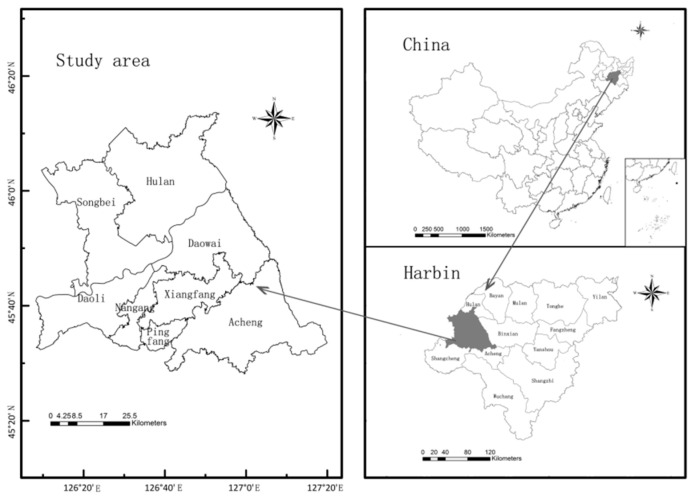
Schematic of study area.

**Figure 2 ijerph-18-01427-f002:**
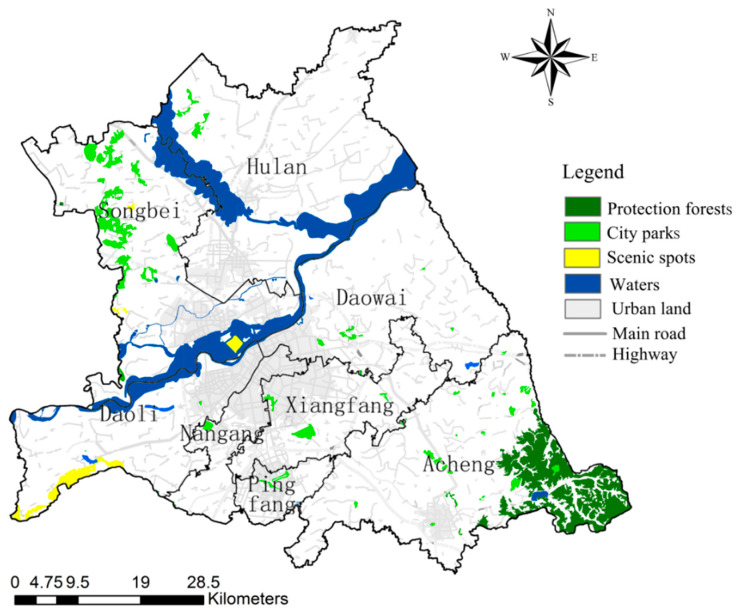
Spatial distribution of ecological elements in the study area.

**Figure 3 ijerph-18-01427-f003:**
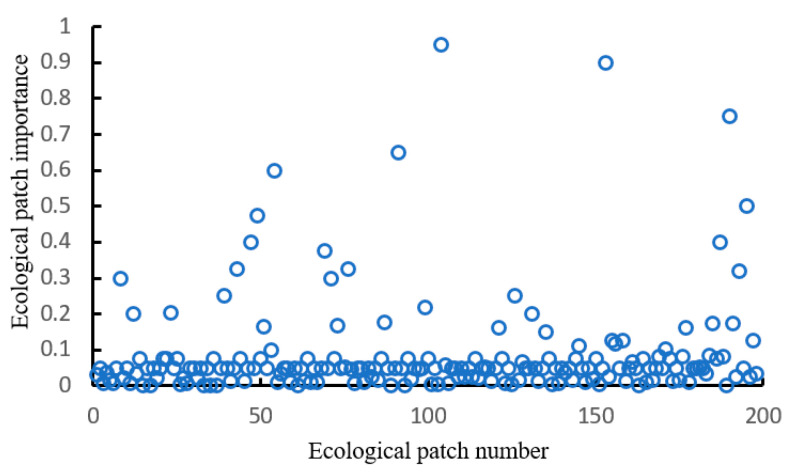
Statistical map of the importance of ecological source sites Q.

**Figure 4 ijerph-18-01427-f004:**
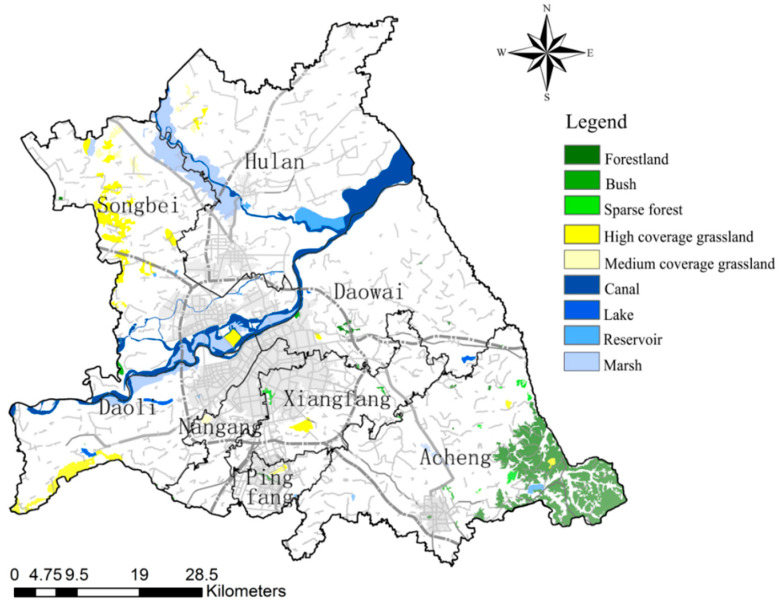
The ecological sources in study the area.

**Figure 5 ijerph-18-01427-f005:**
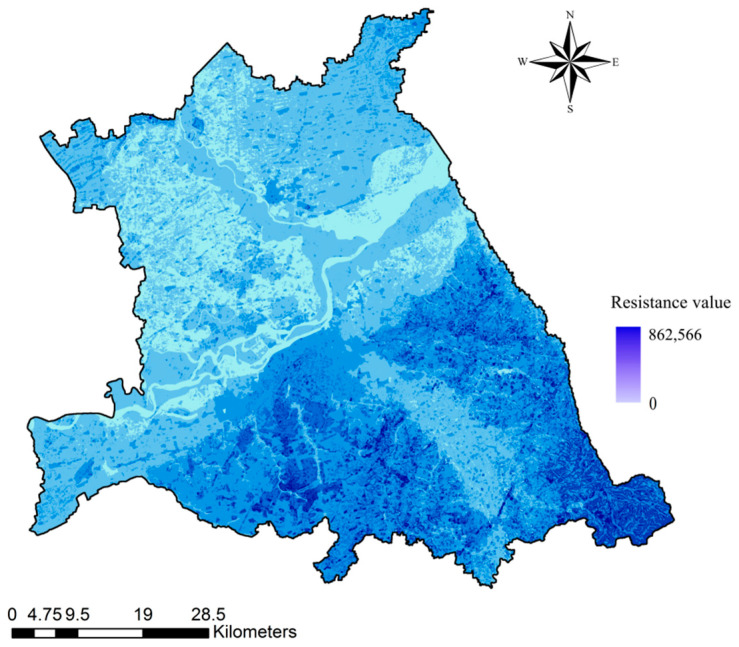
Cost surface.

**Figure 6 ijerph-18-01427-f006:**
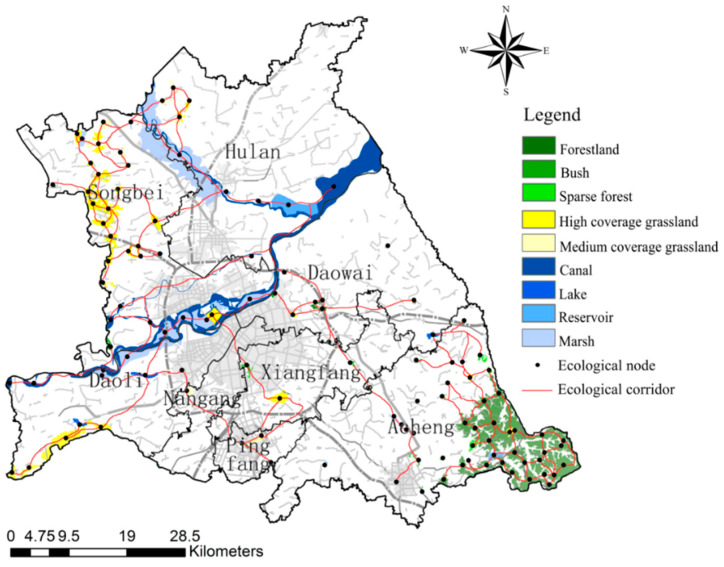
Ecological network in study the area.

**Figure 7 ijerph-18-01427-f007:**
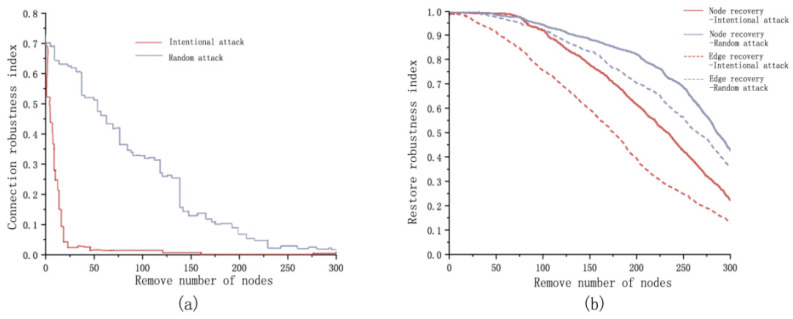
Ecological network robustness analysis before optimization: (**a**) connection robustness analysis of ecological network; (**b**) recovery robustness analysis of ecological network.

**Figure 8 ijerph-18-01427-f008:**
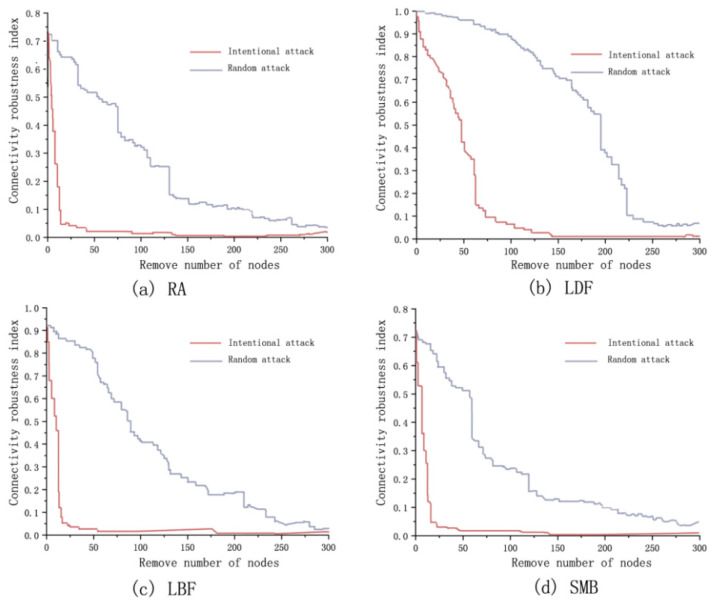
Connection robustness analysis of ecological network with different optimization strategies: (**a**) connection robustness analysis of ecological network under RA; (**b**) connection robustness analysis of ecological network under LDF; (**c**) connection robustness analysis of ecological network under LBF; (**d**) connection robustness analysis of ecological network under SMB.

**Figure 9 ijerph-18-01427-f009:**
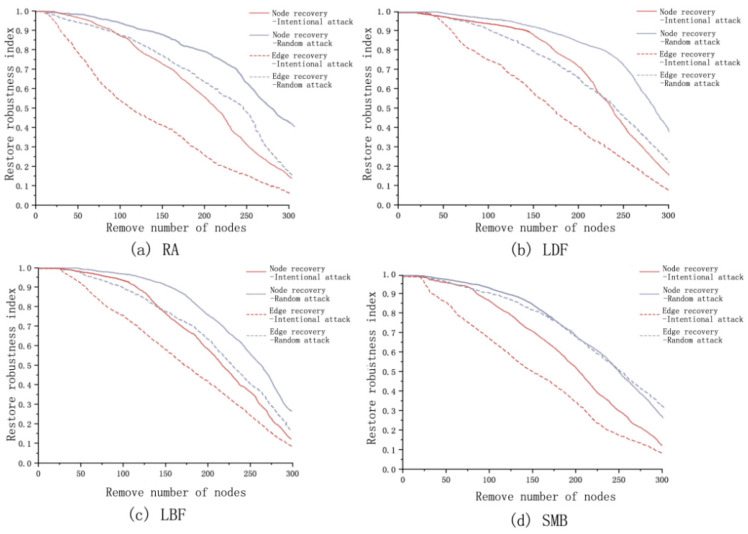
Recovery robustness analysis of ecological network with different optimization strategies: (**a**) recovery robustness analysis of ecological network under RA; (**b**) recovery robustness analysis of ecological network under LDF; (**c**) recovery robustness analysis of ecological network under LBF; (**d**) recovery robustness analysis of ecological network under SMB.

**Figure 10 ijerph-18-01427-f010:**
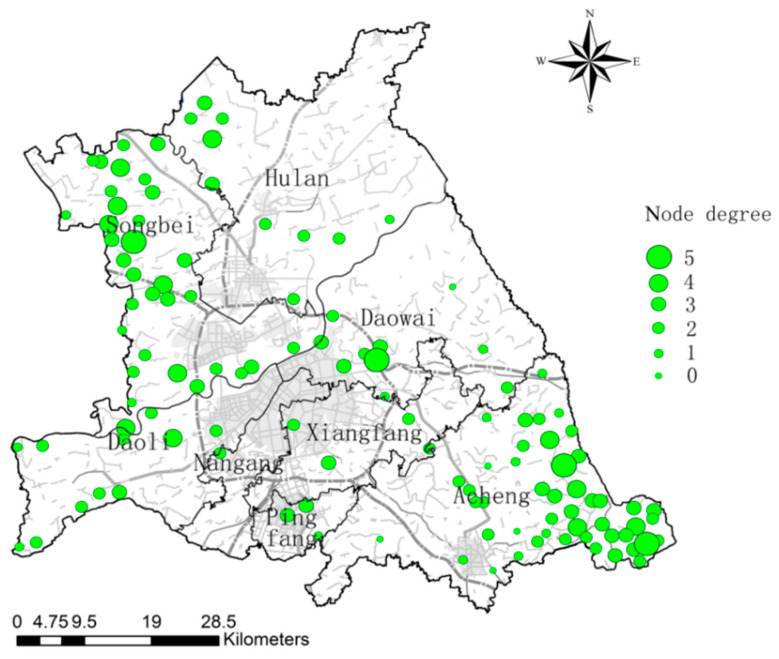
Ecological node degree distribution in the study area before optimization.

**Figure 11 ijerph-18-01427-f011:**
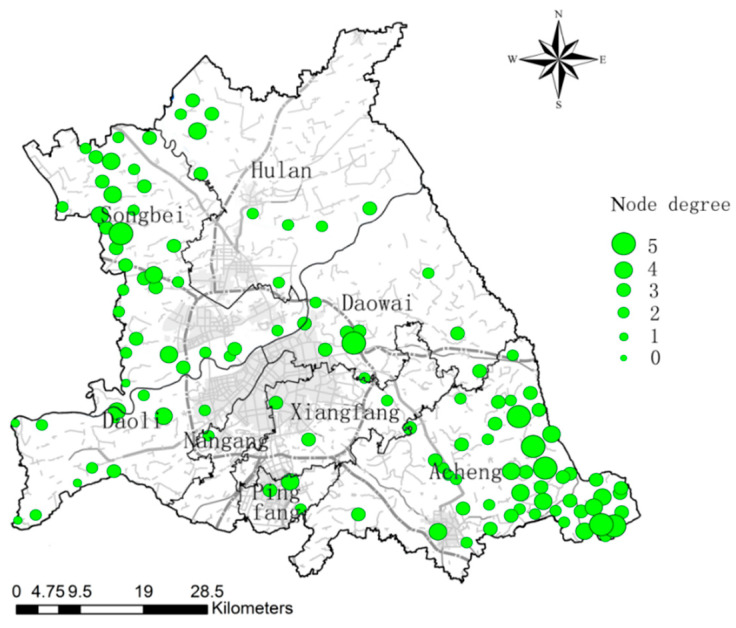
Ecological node degree distribution in the study area after optimization.

**Figure 12 ijerph-18-01427-f012:**
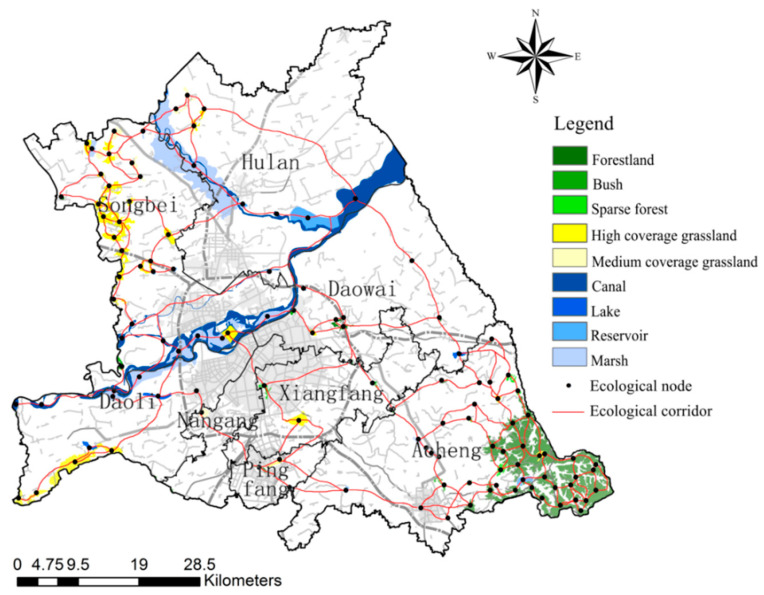
LDF-based ecological network optimization in study area.

**Table 1 ijerph-18-01427-t001:** Landscape pattern index of ecological patch in each district of the study area.

District	NP (Pieces)	CA (ha)	PLAND (%)	LPI (%)	LSI (%)	COHESION (%)
Daoli	19	6951.51	15.6621	8.2965	8.2608	99.3281
Daowai	12	10,946.16	17.7908	15.9749	4.4814	99.6878
Xiangfang	11	798.30	2.3375	1.2647	4.9206	97.3614
Nangang	6	222.48	1.3211	0.9214	2.9100	95.9299
Pingfang	5	269.10	2.9136	1.1518	6.1091	96.7406
Songbei	49	16,097.40	21.8475	14.6009	13.8865	99.5820
Hulan	20	13,858.92	14.5701	13.7796	5.5669	99.7068
Acheng	76	15,855.84	18.7051	16.8907	14.2214	99.7605

## Data Availability

Not applicable.

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
