# Peer review of "Ecological Network Optimization in Urban Central District Based on Complex Network Theory: A Case Study with the Urban Central District of Harbin"

_ijerph, 2021, doi:10.3390/ijerph18041427_

Round 1

Reviewer 1 Report

The text constitutes an interesting approach to the topic of optimization of ecological networks in a large city in North China, Harbin. Specifically, the central district of the city is analyzed. The focus of the work is purely technical, based on a remarkable management of cartographic sources on land use and a very sophisticated level of quantification. The introduction is good. It highlights the interest of the work and its contribution. It is well built and documented. Also the contribution of materials and the method is adequate, with a very prominent analytical and documentary base. However, in this part of the article a great lack of it can already be appreciated: a quantitative study and of land use, completely forgetting the space where it is being carried out. There is no reference, or analysis, that we are facing a densified center of a large city. Urban complexity and the people who live in that city are the great absent from an investigation, which by not establishing a differentiation from the urban is meaningless. The work cannot be published in its current form, until an in-depth study and characterization of the urban fact in the center of a large city is introduced. In this regard, the study area section is insufficient, descriptive and does not provide anything correlated with the research. There is nothing social and at work about a city or ecological networks. This is absolutely necessary. The entire results and discussion section shows this lack. In the same way, the conclusions constitute a cold enumeration of three great results obtained on a city without people, without social interactions. Major revisions are recommended in the direction of the change of analysis model indicated so that the article reaches the necessary quality for its publication.

Author Response

Dear Reviewer:
Thank you very much for your comments on my manuscript entitled “Ecological Network Optimization in Urban Central District Based on Complex Networks Theory: A Case Study with the Urban Central District of Harbin”.(ID:ijerph-1069613). Those comments are all valuable and very helpful for revising and improving our paper, as well as the important guiding significance to our researches. We have studied comments carefully and have made correction which I hope meet with approval. Revised portion are marked in red in the paper. The main corrections in the paper and the responds to the comments are as following:
Responds to the comments of the reviewer:
1. Response to comment: (However, in this part of the article a great lack of it can already be appreciated: a quantitative study and of land use, completely forgetting the space where it is being carried out. There is no reference, or analysis, that we are facing a densified center of a large city. )
Response: Considering the reviewer’s suggestion, I have re-written this part according to the reviewer’s suggestion. The quantitative analysis of the ecological patch characteristics of the study area was removed and replaced by a specific quantitative analysis of the ecological patch characteristics of each jurisdiction and a specific analysis of the ecological status of each jurisdiction according to its main function in the urban master plan (Lines 301-353). In addition, according to the reviewer's opinion, to highlight the urbanization characteristics of the study area in the article, and for this reason, the images in the whole text were completely corrected.

  1. Response to comment: (Urban complexity and the people who live in that city are the great absent from an investigation, which by not establishing a differentiation from the urban is meaningless.In this regard, the study area section is insufficient, descriptive and does not provide anything correlated with the research. There is nothing social and at work about a city or ecological networks. This is absolutely necessary. )
    Response: According to the comment of the reviewer,I added information on the direction of urban planning development in the study area, the current situation of population urbanization, the main problems of urbanization, and the distribution of various ecological factors in each jurisdiction and the current problems they face in the study area section(Lines 136-174). 
    3. Response to comment: (The entire results and discussion section shows this lack. In the same way, the conclusions constitute a cold enumeration of three great results obtained on a city without people, without social interactions. )
    Response: I have made correction according to the Reviewer’s comments. Combined with the correction results of the above comments of the review teacher, the part of the summary and the first conclusion are rewritten.

Special thanks to you for your good comments. 
Other changes: 
   1. Lines 61-64, 67-69,71-79, 82-88 were added. In this section, I mainly added     the theoretical background and the current state of research applications of         complex networks.

  1. Lines94-103. The limitations of existing complex network theory to optimize ecological network studies have been added.
  2. Lines104-110. Reinforced the novelty point of the study
  3. 4.Added figure2 and 11, Lines 511-51 To help illustrate the distribution of ecological elements in the study area and the optimization effect of the LDF optimization strategy.
  4. Lines540-545,293-299. The insights of urbanization process for landscape ecological network planning, the limitations of complex network theory for optimizing landscape ecological networks in this study have been added, and the future research direction of complex network theory for optimizing landscape ecological networks has been pointed out.

    I tried my best to improve the manuscript and made some changes in the manuscript. These changes will not influence the content and framework of the paper. And here I did not list the changes but marked in red in revised paper.

I appreciate for Reviewers’ warm work earnestly, and hope that the correction will meet with approval.
Once again, thank you very much for your comments and suggestions.

Reviewer 2 Report

After reading the manuscript, I have some feedback to the authors for enhancing the manuscript:

  • Ecological network optimisation based on complex network theory was carried out to study Harbin, China. Over the years, landscape pattern analysis has been done extensively and optimisation of ecological network using other approaches is done in Chinese cities (e.g. Beijing-Tianjian-Hebei metropolitan region, Hu et al. 2018; and Tianjin City, Zhao et al. 2019). The authors contended that optimisation of ecological networks based on complex network theory is one of the frontier directions in the field, why? Has the theory been applied to optimise ecological network in the other studies and outside China? The authors are suggested to provide more background of complex network theory in ecological network studies. As a whole, the introduction should provide the study background in more detail and more precisely. The authors should also state more clearly the novelty of this study, and explain why Harbin is chosen for study.
  • The findings were inadequately discussed. It was said that Harbin was chosen for study due to “increasingly more serious ecological and environmental problems”. What are the ecological and environmental problems facing Harbin? From the results, the authors reported that some of the easily identifiable ecological corridors were blocked by urban construction. What are the implications of this study on urban planning? What are the limitations when the complex networks theory is applied to optimise ecological networks?

  • The reviewer has a feel that the manuscript is not ready for publication. The authors have not done the proofreading carefully - some sentences are incomplete (e.g. lines 105-109).

Author Response

Dear Reviewer:
Thank you very much for your comments on my manuscript entitled “Ecological Network Optimization in Urban Central District Based on Complex Networks Theory: A Case Study with the Urban Central District of Harbin”.(ID:ijerph-1069613). Those comments are all valuable and very helpful for revising and improving our paper, as well as the important guiding significance to our researches. I have studied comments carefully and have made correction which I hope meet with approval. Revised portion are marked in red in the paper. The main corrections in the paper and the responds to the comments are as following:
Responds to the comments of the reviewer:
1. Response to comment: (Over the years, landscape pattern analysis has been done extensively and optimisation of ecological network using other approaches is done in Chinese cities (e.g. Beijing-Tianjian-Hebei metropolitan region, Hu et al. 2018; and Tianjin City, Zhao et al. 2019). The authors contended that optimisation of ecological networks based on complex network theory is one of the frontier directions in the field, why?)

Response: I have made correction according to the reviewer’s comments, additions and corrections have been made to this part of the expression, and additional citations have been added(lines 61-64).
2. Response to comment: ( Has the theory been applied to optimise ecological network in the other studies and outside China? The authors are suggested to provide more background of complex network theory in ecological network studies. As a whole, the introduction should provide the study background in more detail and more precisely.)

Response: Considering the reviewer’s suggestion, I have added the purpose and significance of theoretical complex network theory ((lines 67-69), the current situation and main directions of its application in various fields (lines 70-79), and examples of specific research cases of complex network optimization of landscape ecological networks that are representative at the present stage (lines. 81-88). The shortcomings of previous studies on the application of complex network theory to optimize ecological networks and the significance of choosing urban centers as the research object (lines 93-103).

  1. 3. Response to comment: (The authors should also state more clearly the novelty of this study, and explain why Harbin is chosen for study.The findings were inadequately discussed. It was said that Harbin was chosen for study due to “increasingly more serious ecological and environmental problems”. What are the ecological and environmental problems facing Harbin?)

Response: I have re-written this part according to the reviewer’s suggestion, the novelty of this study(lines104-110). the reasons for choosing the central urban area of Harbin as the research area(lines 142-152).
4. Response to comment: (From the results, the authors reported that some of the easily identifiable ecological corridors were blocked by urban construction. What are the implications of this study on urban planning?)
Response: As reviewer suggested, the implications of this study on urban planning were added(Lines 526-535).

  1. 5. Response to comment: (What are the limitations when the complex networks theory is applied to optimise ecological networks?)
    Response:It is really true as reviewer suggested, the limitations and possible directions for future research when applying complex network theory to optimize ecological networks have been added at the end of the article.(Lines 540-545,293-299).
  2. 6. Response to comment: (The reviewer has a feel that the manuscript is not ready for publication. The authors have not done the proofreading carefully - some sentences are incomplete (e.g. lines 105-109).)
    Response:I am very sorry for my incorrect writing, I have carefully proofread the writing of the article. Due to my limited English level, I sincerely hope that the reviewer can help to point out any errors that may have appeared in the article.

Special thanks to you for your good comments. 
Other changes: 
1. Lines 136-174. Added on the layout of urban planning in the study area, the spatial distribution of various ecological factors and the current situation of the problem.

  1. Lines301-353. The original quantitative description of the status quo of ecological patches in the study area has beenrefined to the quantitative description of ecological patches in each jurisdiction, so as to further deepen the understanding of the spatial status of ecological elements of urbanization in the study area.
  2. 3.Added figure2 and 11, lines 511-51 To help illustrate the distribution of ecological elements in the study area and the optimization effect of the LDF optimization strategy.
  3. 4.Lines547-55 The first conclusion re-summarized the quantitative research content of ecological plaque in each jurisdiction.
  4. 5.The previous general description of the direction, such as: southwest or northwest, has been refined to specific jurisdictions.

I tried my best to improve the manuscript and made some changes in the manuscript. These changes will not influence the content and framework of the paper. And here I did not list the changes but marked in red in revised paper.

I appreciate for Reviewers’ warm work earnestly, and hope that the correction will meet with approval.
Once again, thank you very much for your comments and suggestions.

Round 2

Reviewer 1 Report

The authors have assumed the changes in depth that were suggested to them. Therefore the new version they present is considered satisfactory. The article must now be published.

Author Response

Dear Reviewer:
Thank you very much for your comments on my manuscript entitled “Ecological Network Optimization in Urban Central District Based on Complex Networks Theory: A Case Study with the Urban Central District of Harbin”.(ID:ijerph-1069613). Thank you very much for your kind work and consideration of our paper. On behalf of my co-authors, I would like to express my sincere gratitude to you. The main corrections in the paper and the responds to the comments are as following:
1.lines 561-568, the limitations of complex network theory for optimization and current research were explained in more detail.

  1. lines 541-554, the implications of this study on urban planningwas added.
    3. lines 152-167, To what extent has the Harbin city government considered and practiced "urban ecological planning" was added .
  2. lines 555-562, how the current study may bring about changes to "urban ecological planning" in Harbin and other Chinese provinces/cities was further described.

     These changes will not influence the content and framework of the paper. And here I did not list the changes but marked in red in revised paper.

I appreciate for Reviewers’ warm work earnestly, and hope that the correction will meet with approval.
Once again, thank you very much for your comments and suggestions.

Reviewer 2 Report

After reading the revised version and considering authors' responses, I agree that now the article is more ready for consideration of acceptance. I also agree that the authors enhanced the article by providing more background in the introduction, the research design is appropriate, and the methodology has been adequately described. The findings are more discussed. The article is potentially significant. Yet the authors are advised to consider the following points/comments before publication:

(1) The limitations (of complex networks theory to optimise and the current study) are still unclear after considering authors' amendments at lines 540-545 and 293-299; the implications of this study on urban planning were not found in lines 526-535 - there was no change/addition at all?

(2) To what extent has the Harbin city government considered and practiced "urban ecological planning", and how the current study may bring about changes to "urban ecological planning" in Harbin and other Chinese provinces/cities?

Regarding the language and style, the authors are suggested to take note of the followings:

(i) Mistakes such as Harbin Harbin (line 105) and network network (line 296) are spotted. Is it due to the journal submission system? Anyway, it is important for the authors to know it is their duties to proof-read the paper.

(ii) Must check the rules of using acronyms and make corrections accordingly;

(iii) The language and style should be further improved. English editing is needed in this case.

(iv) What do plaque and jurisdiction mean?

(v) The authors should confirm with the editorial office the proper way of in-text citation in MDPI journals.

Author Response

Dear Reviewer:
Thank you very much for your comments on my manuscript entitled “Ecological Network Optimization in Urban Central District Based on Complex Networks Theory: A Case Study with the Urban Central District of Harbin”.(ID:ijerph-1069613). Thank you very much for your kind work and consideration of our paper. On behalf of my co-authors, I would like to express my sincere gratitude to you. The main corrections in the paper and the responds to the comments are as following:

  1. Response to comment: (The limitations (of complex networks theory to optimise and the current study) are still unclear after considering authors' amendments at lines 540-545 and 293-299; the implications of this study on urban planning were not found in lines 526-535 - there was no change/addition at all?)

Response: We have re-written this part according to the Reviewer’s suggestion. The limitations of complex network theory for optimization and current research were further described at lines 561-568, and the implications of this study on urban planning were added at lines 540-553.  

  1. Response to comment: (To what extent has the Harbin city government considered and practiced "urban ecological planning", and how the current study may bring about changes to "urban ecological planning" in Harbin and other Chinese provinces/cities?)

Response: We have made correction according to the Reviewer’s comments. To what extent has the Harbin city government considered and practiced "urban ecological planning" was explained  at lines 152-167, and how the current study may bring about changes to "urban ecological planning" in Harbin and other Chinese provinces/cities added at lines 554-561.

  1. Response to comment: (Regarding the language and style, the authors are suggested to take note ofthe followings(i) Mistakes such as Harbin Harbin (line 105) and network network (line 296) are spotted. Is it due to the journal submission system? Anyway, it is important for the authors to know it is their duties to proof-read the paper.(ii) Must check the rules of using acronyms and make corrections accordingly;(iii) The language and style should be further improved. English editing is needed in this case.(iv) What do plaque and jurisdiction mean?(v) The authors should confirm with the editorial office the proper way of in-text citation in MDPI journals.)

Response: (i) Corrections have been made. (ii) More literature has been consulted for verification. (iii) Due to time, a native English speaker has helped to improve the article, and if there are problems afterwards, we will not hesitate to seek help from the English editor. (iv) The terminology has been standardized in the article. (v) It has been confirmed and corrected.

I tried my best to improve the manuscript and made some changes in the manuscript. These changes will not influence the content and framework of the paper. And here I did not list the changes but marked in red in revised paper.

I appreciate for Reviewers’ warm work earnestly, and hope that the correction will meet with approval.
Once again, thank you very much for your comments and suggestions.
